# Safety Guarantees for Neural Network Dynamic Systems via Stochastic Barrier Functions

**Rayan Mazouz**[1*], **Karan Muvvala**[1*], **Akash Ratheesh**[1], **Luca Laurenti**[2], **Morteza Lahijanian**[1]

[1] Department of Aerospace Engineering Sciences, University of Colorado Boulder, USA
[2] Delft Center for Systems and Control, Delft University of Technology, The Netherlands
{rayan.mazouz,karan.muvvala,akash.ratheesh,morteza.lahijanian}@colorado.edu
{l.laurenti}@tudelft.nl
* Equal Contribution

## Abstract

Neural Networks (NNs) have been successfully employed to represent the state evolution of complex dynamical systems. Such models, referred to as NN dynamic models (NNDMs), use iterative noisy predictions of NN to estimate a distribution of system trajectories over time. Despite their accuracy, safety analysis of NNDMs is known to be a challenging problem and remains largely unexplored. To address this issue, in this paper, we introduce a method of providing safety guarantees for NNDMs. Our approach is based on stochastic barrier functions, whose relation with safety are analogous to that of Lyapunov functions with stability. We first show a method of synthesizing stochastic barrier functions for NNDMs via a convex optimization problem, which in turn provides a lower bound on the system's safety probability. A key step in our method is the employment of the recent convex approximation results for NNs to find piece-wise linear bounds, which allow the formulation of the barrier function synthesis problem as a sum-of-squares optimization program. If the obtained safety probability is above the desired threshold, the system is certified. Otherwise, we introduce a method of generating controls for the system that robustly minimize the unsafety probability in a minimally-invasive manner. We exploit the convexity property of the barrier function to formulate the optimal control synthesis problem as a linear program. Experimental results illustrate the efficacy of the method. Namely, they show that the method can scale to multi-dimensional NNDMs with multiple layers and hundreds of neurons per layer, and that the controller can significantly improve the safety probability.

## 1   Introduction

Safety is a major concern for autonomous dynamical systems, especially in *safety-critical* applications. Examples include UAVs, autonomous cars, and surgical robots. To this end, various methods are developed to provide safety guarantees for such systems [1–3]. Most of these methods, however, assume (simple) analytical models, which are often unavailable or too expensive to obtain due to complexity (nonlinearities) in real-world systems [4]. This has given rise to recent efforts to harvest the representational power of *neural networks* (NNs) to predict how the state of a system evolves over time, referred to as *NN dynamic models* (NNDMs) [5, 6]. Despite their efficient and accurate representation of realistic dynamics, it has been difficult to make formal claims about the behavior of NNDMs over time. This poses a major challenge in safety-critical domains: *how to provide safety guarantees for NNDMs?*

In this paper, we propose a framework to provide safety certificates for NNDMs. The approach is based on stochastic barrier functions [7, 8], whose role for safety is analogous to that of Lyapunov functions for stability, and which can guarantee forward invariance of a safe set. Specifically, we

36th Conference on Neural Information Processing Systems (NeurIPS 2022).

show a method of synthesizing barrier functions for NNDMs via an efficient convex optimization method, which in turn provides a lower bound on the safety probability of the system trajectories. The benefit of our approach is that it does not require unfolding of the NNDM to compute this probability; instead, we base our reasoning on non-negative super-martingale inequalities to bound the probability of leaving the safe set. The key to this is to utilize the recent convex approximation results for NNs to obtain piece-wise linear bounds, and a formulation of the optimization problem that enables the use of efficient solvers. If the obtained safety probability is above a desired threshold, the system can be certified. For the cases that such safety certificates cannot be provided, we introduce a method of generating controls for the system, which robustly maximizes the safety probability in a minimally-invasive manner. We exploit the convexity property of the barrier function to formulate the control synthesis problem as a linear program. Experimental results on various NNDMs trained using state-of-the-art RL techniques [9–11] illustrate the efficacy of the method. Namely, they show that the method can scale to multi-dimensional models with various layers and hundreds of neurons per layer, and that the controller can significantly improve the safety probability.

In summary, this paper makes the following main contributions: (i) a method of computing a lower bound for the safety probability of NNDMs via stochastic barrier functions and efficient nonlinear optimization, thereby providing safety guarantees, (ii) a correct-by-construction control synthesis framework for NNDMs to ensure safety while being *minimally-invasive*, i.e., intervening only when needed, (iii) demonstration of the efficacy and scalability of the framework in benchmarks on various NNDMs with multiple architectures, i.e., multiple hidden layers and hundreds of neurons per layer.

**Related Work** In recent years, NNs have been successfully employed to model complex dynamical systems [5, 6]. Such models use iterative noisy predictions of an NN to estimate the distribution of the system over time. They provide means of predicting state evolution of the systems that have no analytical form, e.g., soft robotics [12]. In addition, because of their ability to quickly unfold trajectories, NNDMs have been used to enhance controller training in Reinforcement Learning (RL) frameworks [9, 11]. In these works, specifically, a NNDM of the system is trained in closed-loop with an NN controller, which can be concatenated in a single NN representing the dynamics of the closed-loop system. Despite their popularity and increased usage, little work has gone to safety analysis of NNDMs. This paper aims to fill this gap.

Barrier functions provide a formal methodology to prove safety of dynamical systems [13, 14]. The state-of-the-art to find stochastic barrier functions is arguably to rely on Sum-of-Squares (SOS) optimization [8]. Barrier certificates obtained via SOS optimization have been studied for deterministic, uncontrolled nonlinear systems [15], and control-affine stochastic systems [16, 8], as well as for hybrid models [13, 14]. In addition, the control synthesis formulation in this paper relaxes the limitations of the existing SOS methods, which generally leads to a non-convex optimization problem [8], by reducing it to a simple linear program.

Many recent studies have focused on the verification of NNs. However, most of them only aim to provide certification guarantees against adversarial examples [17], with approaches ranging from SMT [18], and convex relaxations [19] to linear programming [20]. Such methods, because of their local nature, cannot be directly employed to perform safety analysis on the entire state space and over the trajectories as required for NNDMs. Another line of work focuses on infinite time horizon safety of Bayesian neural networks [21], or the verification of neural ODEs with stochastic guarantees [22]. They consider neural network controllers with known dynamics [23]. Other methods have been employed for trajectory-level properties of NNs based on solution of the Bellman equations [24], formal (finite) abstractions [25], and mixed integer programming [26], which do not support noisy dynamics. In contrast, our approach supports additive noise and only requires to check static properties on the NN by showing that there exists a function whose composition with the NN forms a stochastic barrier function.

## 2    Problem Formulation

In this work, we are interested in providing safety guarantees for dynamical systems that are described by NNs. Specifically, we consider the following discrete-time stochastic process whose time evolution is given by iterative predictions of an NN

$$\mathbf{x}_{k+1} = f^w(\mathbf{x}_k) + \mathbf{v}_k, \qquad k \in \mathbb{N}, \quad \mathbf{x}_k, \mathbf{v}_k \in \mathbb{R}^n, \tag{1}$$

where $f^w$ is a trained, fully-connected, feed-forward NN with general continuous activation functions, and $w$ represents the maximum likelihood weights. Term $\mathbf{v}_k$ is a random variable modelling an additive noise of stationary Gaussian distribution with zero mean and covariance matrix $R \in \mathbb{R}^{n \times n}$, i.e., the probability density function of $\mathbf{v}_k$ is $\mathcal{N}(\bar{x} \mid 0, R)$.

**Remark 1.** *System* (1) *can be viewed as a closed-loop system, where $f^w$ is the composition of a NNDM (that represents the open-loop system) with an NN controller. NNDMs such as the one in System* (1) *are increasingly employed in robotics for both model representation and NN controller training. These models are often obtained by state-of-the-art model-based RL techniques (e.g., [9–11]) or perturbations (step response) of the actual system (e.g., [12]).*

The evolution of System (1) can be characterized by the predictive posterior distribution $p_{\mathbf{x}}(\bar{x} \mid x)$, which describes the probability density of $\mathbf{x}_{k+1}$ when $\mathbf{x}_k = x \in \mathbb{R}^n$. This distribution is induced by noise $\mathbf{v}_k$ and hence is Gaussian with mean $f^w(x)$ and covariance $R$, i.e., $p_{\mathbf{x}}(\bar{x} \mid x) = \mathcal{N}(\bar{x} \mid f^w(x), R)$. For a subset of states $X \subseteq \mathbb{R}^n$ and (initial) state $x \in \mathbb{R}^n$, we call $T(X \mid x) = \int_X p_{\mathbf{x}}(\bar{x} \mid x) d\bar{x}$ the *stochastic kernel* of System (1). From the definition of $T$ it follows that, for a given initial condition $\mathbf{x}_0 = x_0 \in \mathbb{R}^n$, $\mathbf{x}_k$ is a Markov process with a well-defined probability measure Pr uniquely generated by the stochastic kernel $T$ [27, Proposition 7.45]. For $X_0, X_k \subseteq \mathbb{R}^n$, this probability measure is inductively defined as

$$\Pr[\mathbf{x}_0 \in X_0] = \mathbf{1}_{X_0}(x_0), \qquad \Pr[\mathbf{x}_k \in X_k \mid \mathbf{x}_{k-1} = x] = T(X_k \mid x),$$

where indicator function $\mathbf{1}_{X_0}(x_0) = 1$ if $x_0 \in X_0$, otherwise 0. The definition of Pr allows one to make probabilistic statements over the trajectories of System (1). In this work, we are particularly interested in two main problems: safety certification and safe controller synthesis for System (1).

**Problem 1** (Safety Certificate). *Let $X_s \subset \mathbb{R}^n$ and $X_0 \subseteq X_s$ be basic closed semi-algebraic sets representing respectively the safe set and the set of initial states. Denote the probability that the system initialized in $X_0$ remains in the safe set in the next $N \in \mathbb{N}_{\geq 0}$ steps by*

$$P_s(X_s, X_0, N) = Pr[\forall x_0 \in X_0, \forall k \leq N, \mathbf{x}_k \in X_s]. \tag{2}$$

*Then, given threshold $\delta_s \in [0, 1]$, provide a safety certificate for System* (1) *by proving that $P_s(X_s, X_0, N) \geq \delta_s$.*

Problem 1 seeks to compute the probability that $\mathbf{x}_k$ remains within a safe set, e.g., it avoids obstacles or any undesirable states. Computation of this probability is particularly challenging because System (1) is stochastic and NN ($f^w$) is generally a non-convex function. The assumption that $X_s$ and $X_0$ are basic closed semi-algebraic sets is not limiting, as these sets are defined as an intersection of a finite number of polynomial inequalities and, for instance, include convex polytopes and spectrahedra [28].

In the case that a safety certificate cannot be generated, we are interested in synthesizing a controller that *by-design* guarantees safety. To this end, we extend System (1) with an affine input

$$\mathbf{x}_{k+1} = f^w(\mathbf{x}_k) + g\mathbf{u}_k + \mathbf{v}_k, \tag{3}$$

where $g : \mathbb{R}^c \to \mathbb{R}^n$ is given, $\mathbf{u}_k \in U \subset \mathbb{R}^c$ is a control action, and $U$ is bounded. Our goal is to synthesize a feedback controller $\pi : \mathbb{R}^n \to U$ such that $\mathbf{u}_k = \pi(\mathbf{x}_k)$ guarantees safety of System (3).

**Problem 2** (Safe Control Synthesis). *Let $X_s \subset \mathbb{R}^n$ and $X_0 \subseteq X_s$ be basic closed semi-algebraic sets representing respectively the safe set and the set of initial states, and $N \in \mathbb{N}_{>0}$ a time horizon. Then, given threshold $\delta_s$, synthesize a feedback controller $\pi$ such that $P_s(X_s, X_0, N, \pi) \geq \delta_s$, where $P_s(X_s, X_0, N, \pi)$ is the safety probability obtained for control-affine System* (3).

**Remark 2.** *There may exist many possible controllers that are solutions to Problem 2. Since System* (1) *represents a closed-loop system with a given controller, one may want to specifically synthesize a safety controller that is "minimally-invasive," i.e., it minimally intervenes to guarantee safety. In Section 5, we elaborate on this notion and introduce methods of generating such controllers.*

**Overall Approach**   Our approaches to Problems 1 and 2 are based on (stochastic) barrier functions. Specifically, we aim to synthesize barrier functions that can serve as a safety certificate by providing a lower bound for $P_s$. To achieve this, we rely on recent results for local convex relaxation of an NN to find piece-wise linear over- and under-approximations of $f^w$. This allows us to formulate the problem of finding a stochastic barrier function for System (1) as an SOS optimization problem.

Our (minimally-invasive) control synthesis method (Problem 2) focuses on injecting controls to System (3) only in the regions of the state space that potentially contribute to unsafety of the system. To this end, we exploit two main properties of the barrier function: the required (super-) martingale property and the convexity property of SOS polynomials. The former allows us to design a method of identifying the "unsafe" regions, and the latter enables us to formulate a Linear Programming (LP) optimization problem to find controllers that robustly minimize the probability of unsafety.

**Remark 3.** *For ease of presentation, $g$ in (3) is taken to be a constant, however, we note that the proposed approach can also handle $g$ being a continuous function of $x$, or even a NN. For the latter, we introduce an interval bound propagation approach in Section 4.1 that allows the computation of upper and lower bounds on $g(x)$, which in turn can be incorporated in the LP optimization formulation in Section 5 for control synthesis (Problem 2) in a straightforward manner.*

# 3 Preliminaries

In this section, we provide a brief background on stochastic barrier functions and non-negative polynomials, which are central components of our framework.

## 3.1 Stochastic Barrier Functions

Consider stochastic process $\mathbf{x}_{k+1} = F(\mathbf{x}_k, \mathbf{v}_k)$, where $\mathbf{x}_k \in X \subseteq \mathbb{R}^n$, $\mathbf{v}_k \in V \subseteq \mathbb{R}^m$ is a well-defined stochastic process, and $F : X \times V \to X$ is a locally Lipschitz continuous function. Then, given initial set $X_0 \subset X$, safe set $X_{\mathrm{s}} \subseteq X$, and unsafe set $X_{\mathrm{u}} \subset X$, a twice differentiable function $B : X \to \mathbb{R}$ is called a *stochastic barrier function* or simply barrier function if the following conditions hold for $\alpha \geq 1$ and $\beta, \eta \in [0, 1]$:

$$B(x) \geq 0 \qquad\qquad \forall x \in X \qquad\qquad (4\mathrm{a})$$
$$B(x) \leq \eta \qquad\qquad \forall x \in X_0 \qquad\qquad (4\mathrm{b})$$
$$B(x) \geq 1 \qquad\qquad \forall x \in X_{\mathrm{u}} \qquad\qquad (4\mathrm{c})$$
$$E[B(F(x,v)) \mid x] \leq B(x)/\alpha + \beta \qquad \forall x \in X_{\mathrm{s}} \qquad\qquad (4\mathrm{d})$$

If such a $B$ exists, then for any $N \in \mathbb{N}_{\geq 0}$, it follows, assuming $\alpha = 1$ without loss of generality, that

$$\Pr[\forall k \leq N, \ \mathbf{x}_k \in X_{\mathrm{s}} \mid \mathbf{x}_0 \in X_0] \geq 1 - (\eta + \beta N). \qquad\qquad (5)$$

Intuitively, Conditions (4a)-(4d) allow one to use non-negative super-martingale inequalities to bound the probability that the system reaches the unsafe region. This results in the probability bound in (5) (see [29, Chapter 3, Theorem 3]). A major benefit of using barrier functions for safety analysis is that while Conditions (4b)-(4d) are static properties, they allow one to make probabilistic statements on the time evolution of the system without the need to evolve the system over time.

## 3.2 Non-negative Polynomials

Synthesis of barrier functions can be formulated as a nonlinear optimization problem. To solve it efficiently via convex programming, we formulate the problem using SOS polynomials, which is a sufficient condition for polynomial non-negativity [30]. This formulation allows the use of efficient semidefinite programming solvers, which have polynomial worst-case complexity [31].

**Definition 1** (SOS Polynomial)**.** *A multivariate polynomial $\lambda(x)$ is a Sum-Of-Squares (SOS) for $x \in \mathbb{R}^n$ if there exists some polynomials $\lambda_i$, $i = 1, \ldots, r$, for some $r \in \mathbb{N}$, such that $\lambda(x) = \sum_{i=1}^{r} \lambda_i^2(x)$. If $\lambda(x)$ is an SOS, then $\lambda(x) \geq 0$ for all $x \in \mathbb{R}^n$. The set of all SOS polynomials is denoted by $\Lambda$.*

Now, consider a basic closed semi-algebraic set $X = \{x \in \mathbb{R}^n \mid h_i(x) \geq 0 \ \forall i \in \{1, \ldots, l\}\}$, which is defined as the intersection of $l$ polynomial ($h_i(x)$) inequalities. Then, Proposition 1 allows one to enforce non-negativity on these semi-algebraic sets.

**Proposition 1** (Putinar's Certificate, [32])**.** *Let $X \subset \mathbb{R}^n$ be a compact basic semi-algebraic set defined as $X = \{x \in \mathbb{R}^n \mid h_i(x) \geq 0 \ \forall i \in \{1, \ldots, l\}\}$, where $h_i(x)$ is a polynomial. Then, polynomial $\gamma(x)$ is non-negative on $X$ if, for some $\lambda_0(x), \lambda_i(x) \in \Lambda$, $\gamma(x) = \lambda_0(x) + \sum_{i=1}^{l} \lambda_i(x)h_i(x)$.*

**Corollary 1.** *Let $\gamma(x)$ be a non-negative polynomial on $X = \{x \in \mathbb{R}^n \mid h_i(x) \geq 0 \quad \forall i \in \{1, \ldots, l\}\}$, and $h(x)$ be an l-dimensional vector of polynomials with $h_i(x)$ as its i-th dimension for all $i \in \{1, \ldots, l\}$. Further, let $\mathcal{L}$ be an l-dimensional vector of SOS polynomials, i.e., the i-th element of $\mathcal{L}$ is $\lambda_i(x) \in \Lambda$. Then, it holds that $\gamma(x) - \mathcal{L}^T h(x) \in \Lambda$.*

# 4 Barrier Functions for Neural Network Dynamical Models

In this section, we introduce a formulation for synthesizing a barrier function that guarantees safety of System (1). Our approach relies on the local convex relaxations of $f^w$, which we use to formulate an SOS optimization problem that generates a valid barrier function for System (1).

## 4.1 Local Convex Relaxation of NNDMs

Our approach uses a convex relaxation of $f^w$. Specifically, we utilize the recent relaxation results for NNs [19] to compute piece-wise linear functions that under- and over-approximate $f^w$. The advantage of this method is that it is computationally efficient and produces tight bounds for small neighborhoods. To this end, we first partition $X_s$ to a finite set of regions $Q = \{q_1, \ldots, q_{|Q|}\}$, where $q_i \subseteq X_s$ for all $i \in \{1, \ldots, |Q|\}$ is a basic (closed) semi-algebraic set. Such a decomposition can be obtained using, e.g., a grid or half-spaces. Then, for every $q \in Q$, we compute linear bounds

$$\underline{f}_q(x) = \underline{A}_q x + \underline{b}_q, \quad \overline{f}_q(x) = \overline{A}_q x + \overline{b}_q \quad \text{such that} \quad \forall x \in q, \quad \underline{f}_q(x) \leq f^w(x) \leq \overline{f}_q(x). \quad (6)$$

The above linear functions can be produced in several ways. One method is to forward propagate the bounds in the input domain of $f^w(x)$ using Interval Bound Propagation (IBP) based approaches [33, 34]. These approaches are efficient but generally lead to loose approximations. Another method is Linear Relaxation based Perturbation Analysis (LiRPA) algorithms, which perform a symbolic back propagation followed by a forward evaluation of the bounds [35, 36, 19]. Such LiRPA-based algorithms produce tighter bounds but can be slow. In our implementation, we use the off-the-shelf toolbox [36] to compute $\overline{f}_q(x)$ and $\underline{f}_q(x)$ in (6) for its trade-off between speed and accuracy.

## 4.2 Barrier Function (Certificate) Synthesis

Recall that a valid barrier function $B(x)$ must satisfy Conditions (4b)-(4d). With the above local linear bounds for $f^w$, Condition (4d) can be redefined over the partitioned regions in $Q$. That is, Condition (4d) can be replaced with the following constraints: for each $q \in Q$,

$$E[B(f_q^w(x) + \mathbf{v}) \mid x] \leq B(x)/\alpha + \beta \quad \text{and} \quad \underline{f}_q(x) \leq f_q^w(x) \leq \overline{f}_q(x) \quad \forall x \in q. \quad (7)$$

In this formulation, $f_q^w(x)$ is a (free) variable that is bounded by two linear functions, avoiding the non-convexity of $f_q^w(x)$. Consequently, an efficient evaluation of the constraint becomes feasible.

**Remark 4.** *Note that there are in fact three constraints in (7) (one super-martingale condition on $B$ and two bounds on $f_q^w(x)$). If they are treated separately, it leads to a conservative barrier function because each constraint needs to hold for all $x \in q$ separately. A less conservative approach is to pose all constraints for the same $x$ at the same time. In Theorem 1, we show how this can be achieved.*

As commonly practiced in the literature, e.g., [15, 8, 37], we restrict our search for a $B(x)$ to the class of polynomial functions. Then, the degree of the polynomial becomes a design parameter. In the following lemma, we show that the degree of the polynomial must be at least 2 for the polynomial to be a valid candidate barrier function for System (1). That is, there does not exist a linear function satisfying Conditions (4b)-(4d) for $\eta < 1$, which implies that the lower bound in (5) is trivially 0.

**Lemma 1.** *For $k \in \mathbb{R}^n$ and $c \in \mathbb{R}$, consider linear function $B(x) = k^T x + c$. Then, $B(x)$ satisfies Conditions (4b)-(4d) iff $k = 0$ and $c = 1$.*

*Proof.* There are two cases to consider: (i) $k = 0$: in this case $B(x) = c$, which satisfies Conditions (4b)-(4d) iff $c = \eta = 1$. (ii) $k \neq 0$: in this case, due to the linearity of $B(x)$, there always exists $x' \in \mathbb{R}^n$ such that $B(x') < 0$, violating Condition (4a). $\qquad\qquad\square$

Having established that the polynomial degree must be at least 2, we further restrict $B(x)$ to SOS polynomials. Theorem 1 shows a formulation of the barrier function synthesis problem as an SOS program, which can be solved in polynomial time. In particular, the theorem shows how to formulate Conditions (4b)-(4c) and (7) as SOS constraints.

**Theorem 1** (NNDM Barrier Certificate). *Consider SOS polynomial function $B(x)$, and safe set $X_{\mathrm{s}} = \{x \in \mathbb{R}^n \mid h_{\mathrm{s}}(x) \geq 0\}$, initial set $X_0 = \{x \in \mathbb{R}^n \mid h_0(x) \geq 0\}$, unsafe set $X_{\mathrm{u}} = \mathbb{R}^n \setminus X_{\mathrm{s}} = \{x \in \mathbb{R}^n \mid h_{\mathrm{u}}(x) \geq 0\}$, and partition region $q = \{x \in \mathbb{R}^n \mid h_q(x) \geq 0\}$ for all $q \in Q$. Let $\mathcal{L}_{\mathrm{s}}(x)$, $\mathcal{L}_0(x)$ and $\mathcal{L}_{\mathrm{u}}(x)$ be vectors of SOS polynomials with the same dimensions as $h_{\mathrm{s}}$, $h_0$, and $h_{\mathrm{u}}$, respectively. Likewise, let $\mathcal{L}_{q,x}(x)$ and $\mathcal{L}_{q,y}(x)$ be vectors of SOS polynomials with the same dimension as $h_q$. Then, a stochastic barrier certificate $B(x)$ for System* (1) *with time horizon $N \in \mathbb{N}_{\geq 0}$ can be obtained by solving the following SOS optimization problem for $\eta, \beta \in [0, 1]$:*

$$\min_{\beta, \eta} \quad \eta + \beta N \qquad \textit{subject to:}$$

$$B(x) \in \Lambda, \tag{8a}$$

$$- B(x) - \mathcal{L}_0^T(x)h_0(x) + \eta \in \Lambda, \tag{8b}$$

$$B(x) - \mathcal{L}_{\mathrm{u}}^T(x)h_{\mathrm{u}}(x) - 1 \in \Lambda, \tag{8c}$$

$$- E[B(y + v) \mid x] + B(x)/\alpha + \beta - \mathcal{L}_{q,x}^T(x)h_q(x) -$$
$$\mathcal{L}_{q,y}^T(x)(\overline{f}_q(x) - y)(y - \underline{f}_q(x)) \in \Lambda \qquad \forall q \in Q \tag{8d}$$

*which guarantees safety probability $P_{\mathrm{s}}(X_{\mathrm{s}}, X_0, N) \geq 1 - (\eta + \beta N)$.*

*Proof.* It suffices to show that if $B$ satisfies Constraints (8a)-(8d), then Conditions (4a)-(4d) are satisfied. As Constraint (8a) guarantees that $B$ is a SOS polynomial, hence non-negative by definition, Condition (4a) holds. By Corollary 1 it holds that if Constraints (8b)-(8c) hold, then $B$ is respectively smaller that $\eta$ in $X_0$ and greater than 1 in $X_u$. What is left to show is that Constraint (8d) guarantees the satisfaction of Condition (4d). This can be done as follows. For every region $q \in Q$, the expectation term in Condition (4d) can be expressed as $E[B(y + v) \mid x]$, where $y$ is bounded by under-approximation $\underline{f}_q(x)$ and over-approximation $\overline{f}_q(x)$ of $f^w(x)$ in System (1) for all $x \in q$. Then by Corollary 1, Constraint (8d) is obtained for each $q$. Note that $B(x)$ is a polynomial, so if $E[B(y + v) \mid x]$ is also a polynomial, the condition is a valid SOS constraint. Given that $y + v$ is linear, $E[B(y + v) \mid x]$ is a sum of monomials in terms of $y$ and expectation moments $\mathbb{E}[v^d]$, where $d \geq 0$. Since, random variable $v$ has a normal distribution, $\mathbb{E}[v^d]$ is constant. Therefore, $E[B(y + v) \mid x]$ is a polynomial only in terms of (components of) $y$. $\qquad \square$

**Remark 5.** *We note that Condition* (8d) *uses linear equations in* (6) *to bound $f^w(x)$ for all $x \in q$. Removing the dependence on $x$, by replacing the equations for $\overline{f}_q(x)$ and $\underline{f}_q(x)$ with their extreme values in region $q$, relaxes the optimization problem and results in a speed-up. It is however an additional over-approximation, and hence the obtained safety probability bound is strictly smaller than the one generated by using the actual linear equations for $\overline{f}_q(x)$ and $\underline{f}_q(x)$. This is the classical "efficiency versus accuracy" trade-off. In our implementation, we compare the two approaches.*

# 5 Minimally-invasive Barrier Controller Synthesis

Here, we focus on Problem 2. The assumption is that a barrier function $B(x)$ is synthesized for System (1) via Theorem 1, which has a safety probability that is lower than threshold $\delta_{\mathrm{s}}$. Hence, our goal is to design a feedback controller for System (3) that guarantees safety. Ideally, this controller is minimal in the interruptions it causes to the system evolution. Below, we first provide a definition for a minimally-invasive controller, and then show how it can be synthesized.

Recall that a feedback controller $\pi : \mathbb{R}^n \to U$ is a function that assigns a control value to each state. Let $X_\pi \subseteq X_{\mathrm{s}}$ be the set of states, to which $\pi$ assigns non-zero control values, i.e., $\pi(x) \neq 0$ for all $x \in X_\pi$. Further, we denote by $V(X)$ the volume of set $X \subset \mathbb{R}^n$. Then, we say a feedback controller $\pi$ is *minimally-invasive* if it minimizes $V(X_\pi)$. Moreover, we say $\pi$ is $\epsilon$-*minimally-invasive* if $V(X_\pi) - \min_{\pi'} V(X_{\pi'}) \leq \epsilon$. In this work, we seek $\epsilon$-*minimally-invasive* controller with an arbitrary small $\epsilon$. To this end, we make the observation that if the discretization of $X_{\mathrm{s}}$ discussed in Section 4.1

is uniform, $V(X_\pi)$ for a given $\pi$ is directly proportional to the number of discrete regions in $Q$ to which $\pi$ applies a non-zero control. We denote by $Q_\pi$ the set of such regions. Then, a feedback controller $\pi$ that minimizes $|Q_\pi|$ is $\epsilon$-*minimally-invasive*, and as the discretization becomes finer, $\epsilon$ monotonically approaches zero. Therefore, given barrier function $B(x)$, our first goal is to identify the minimal number of regions in $Q$ that require a non-zero controller to guarantee safety. Recall that, given $B(x)$, we require the safety probability to be $1 - (\eta + \beta N) \geq \delta_s$. While $\eta$ is related to the value of $B(x)$ in the initial set, $\beta$ is a compensation needed to turn a sub-martingale inequality to a super-martingale inequality (Condition (8d) or (4d)). We can compute this compensation for each region $q \in Q$ by evaluating Condition (8d) or (4d). Let $\beta_q$ denote the computed compensation term for region $q$. Then, region $q$ requires a non-zero controller if $\beta_q > (1 - \delta_s - \eta)/N$.

We now turn our attention to designing a controller that reduces $\beta_q$. In our approach, we tap directly into the convex nature of the SOS polynomial $B(x)$ (SOS-convex). Lemma 2 shows that a controller that drives System (3) to a point closest to the minimum point of $B(x)$, also robustly minimizes $\beta_q$.

**Lemma 2.** *Given $B(x)$, let $x^* = \arg\min_x B(x)$, $x' = f^w(x) + gu + v$ as in System (3), and $\beta_q \geq E[B(x') \mid x] - B(x)$ for all $x$ in region $q \subseteq X_s$. Then, the controller that minimizes $\|\tilde{x}' - x^*\|_1$, where $\tilde{x}' = f^w(x) + gu$, also robustly minimizes $\beta_q$, i.e., minimizes an upper bound of $\beta_q$.*

*Proof.* Given that $B$ is a polynomial and $x'$ is affine in both $u$ and $v$, we can write $E[B(x') \mid x] = B(E[x' \mid x]) + \xi$, where $\xi$, called Jensen's gap [38], is a non-negative polynomial of $x$, $u$, and moments $E[v^d]$, where $2 \leq d \leq m$ is an even integer. This polynomial can always be upper bounded by a constant $\bar{\xi} \geq 0$ for bounded $x$ and $u$. Then, with $\tilde{x}' = E[x']$, the min $\beta_q$ is upper bounded by

$$\min_u \beta_q \leq \min_u \max_{x \in q} \left( B(\tilde{x}' \mid x) - B(x) + \bar{\xi} \right) \leq \min_u \max_{x \in q} B(\tilde{x}' \mid x) - \min_{x \in q} B(x) + \bar{\xi}.$$

The second inequality holds because $B$ is a non-negative function resulting from a SOS optimization, and is thus SOS-convex [39]. Since $B(\tilde{x}')$ monotonically decreases as $\tilde{x}' \to x^*$, $\max_{x \in q} B(\tilde{x}' \mid x)$ is minimized by $\arg\min_u \|f^w(x) + gu - x^*\|_1 \ \forall x \in q$. $\qquad\square$

We note that, in the proof of Lemma 2, the moments $E[v^d]$ appear in every monomial of polynomial $\xi$ (Jensen's gap). Hence, for $v$ with variance less than 1, Jensen's gap is very small and vanishes as the variance becomes smaller, making the controller in the lemma optimal. Furthermore, even though Lemma 2 uses $L_1$ norm for $\|\tilde{x}' - x^*\|_1$, the statement also holds for $L_2$ and $L_\infty$ norms. We specifically use $L_1$ norm to be able to compute the controller via an LP as stated in Theorem 2.

**Theorem 2** (Controller Synthesis)**.** *Consider convex barrier function $B(x)$ and its minimum argument $x^*$ as specified in Lemma 2, and linear bounds $\underline{f}_q(x)$ and $\overline{f}_q(x)$ for $f^w(x)$ in region $q \subseteq \mathbb{R}^n$ as in (6). A controller that robustly minimizes $\beta_q$ for System (3) is the solution to the following LP optimization problem, where $\theta = (\theta_1, \theta_2, \ldots, \theta_n) \in \mathbb{R}^n_{\geq 0}$, and the vector inequality relations are element-wise:*

$$\min_{\theta_i, z', z'', u} \sum_{i=1}^n \theta_i \qquad \textit{subject to:}$$

$$
\begin{array}{ll}
y - x^* \leq \theta & \text{(10a)} \\
x^* - y \leq \theta & \text{(10b)} \\
z = z' - z'' & \text{(10c)}
\end{array}
\qquad
\begin{array}{ll}
y \geq \underline{f}_q(z) + gu & \text{(10d)} \\
y \leq \overline{f}_q(z) + gu & \text{(10e)} \\
z', z'' \geq 0, \ z \in q, \ u \in U & \text{(10f)}
\end{array}
$$

*Proof.* By Lemma 2 it holds that to robustly minimizes $\beta_q$, it suffices to minimize $\|\tilde{x}' - x^*\|_1$. This norm is the sum of $n$ absolute values, each of which can be set to $\theta_i$ with linear Constraints (10a) and (10b). Hence, the minimization of $\|\tilde{x}' - x^*\|_1$ is equivalent to minimization of $\sum_{i=1}^n \theta_i$ subject to (10a) and (10b). The bounds on the dynamics of System (3) can be expressed as linear Conditions (10d) and (10e), each of which has to hold for all $z \in q$, i.e., $z$ is a free variable. To turn the optimization into an LP, $z$ is expressed as the difference of two non-negative decision variables $z'$ and $z''$ in Constraint (10c). $\qquad\square$

**Remark 6.** *Note that the controller obtained via the LP is a vector of scalars for each region $q$. This leads to a feedback controller $\pi$ over the regions $q \in Q$. It is possible to express $u$ as SOS polynomials, which would be a feedback controller over $x$. That could potentially provide a tighter bound for $\beta_q$ at the cost of extra computation since SOS optimization is more expensive than LP.*

## 5.1 Control Synthesis Algorithm

Our framework for computing a barrier certificate as well as minimally-invasive controllers for NNDMs is outlined in Algorithm 1. First, the safe set $X_s$ is uniformly partitioned into a finite set of regions $|Q|$, and bounds of $f^w$ are computed as piece-wise linear functions per Section 4.1. Then, for a given polynomial degree $m \geq 2$, a barrier certificate along with its corresponding safety probability bound are computed in accordance with Theorem 1. If this probability is greater than or equal to $\delta_s$, the NNDM is certified, and the algorithm terminates. Otherwise, it iteratively searches for $\epsilon$-minially-invasive controllers. In each iteration, upper bound $\bar{\eta}$ is set on $\eta$ (initially with its largest possible value $1 - \delta_s$ and then reduced by $\Delta\eta > 0$), and a new barrier function is computed with this bound, where the objective function is minimization of $\beta$ to ensure $\epsilon$-minimally-invasive controllers. Then, for each $q \in Q$, $\beta_q$ is checked against the threshold. If exceeding, a controller is computed for $q$ in accordance with Theorem 2. This process repeats until a satisfactory $P_s$ is obtained or $\bar{\eta} < 0$.

We highlight two important properties of this algorithm: it clearly terminates in finite time, and if $P_s \geq \delta_s$, the synthesized controller is correct-by-construction.

---

**Algorithm 1:** NNDM Controller Synthesis

    **Input** : NN $f^w$, initial set $X_0$, safe set $X_s$, noise covariance $R$, polynomial degree $m$,
              threshold $\delta_s$, and step size $\Delta\eta$
    **Output :** feedback controller $\pi$, and safety probability bound $P_s$

1   $Q, \underline{f}_q, \overline{f}_q \leftarrow$ PARTITIONANDCOMPUTEBOUNDS$(X_s, f^w)$,     $k \leftarrow 0$
2   $\eta, \beta, B(x) \leftarrow$ COMPUTEBARRIERCERTIFICATE$(Q, \underline{f}_q, \overline{f}_q, R, m)$            // Theorem 1
3   **while** $P_s < \delta_s$ *and* $\bar{\eta} > 0$ **do**
4      $\bar{\eta} \leftarrow 1 - (\delta_s - k\Delta\eta)$,    $k \leftarrow k+1$                        // set upper bound on $\eta$
5      $\eta, \beta, B(x) \leftarrow$ COMPUTEBARRIERCERTIFICATE$(Q, \underline{f}_q, \overline{f}_q, R, m, \bar{\eta})$      // Theorem 1
6      **for** $q \in Q$ **do**
7          $\pi \leftarrow (q, 0)$
8          **if** $\beta_q > (1 - \delta_s - \eta)/N$ **then**
9              $\pi \leftarrow (q, u_q) \leftarrow$ COMPUTECONTROL$(\arg\min B(x), q, \underline{f}_q, \overline{f}_q, U)$   // Theorem 2
10            $\beta_q \leftarrow$ UPDATEBETA$(B(x), u_q)$                     // evaluate Condition (8d)
11      $P_s \leftarrow 1 - (\eta + N \max_{q \in Q} \beta_q)$
12 **return** $\pi, P_s$

---

# 6 Case Studies

We demonstrate the efficacy of our framework on several case studies. We first show that our verification method is able to produce non-trivial safety guarantees for NNDMs trained using imitation learning techniques [40] from data gathered by rolling out an RL agent trained with state-of-the-art RL techniques [41]. We then show that our control approach significantly increases the safety probability.

**Models** We trained the *Pendulum* (*2D*), *Cartpole* (*4D*), and *Acrobot* (*6D*) agents from the OpenAI gym environment, as well as *4D* and *5D Husky robot* models. For the OpenAI models, we collected input-output states of the system under an expert controller, either as a look-up table or an NN controller. We picked the best controller available for the agents from the OpenAI Leaderboard [41]. The Husky models were trained to move from one point to another while staying within a lane. We initially trained a NNDM along with an NN controller on the trajectory data generated from PyBullet (physics simulator [42]) as outlined in [9]. We then used these NNs to generate input-output data and trained closed-loop NNDMs as specified above. In total, we trained 10 fully-connected multi-layer perceptron NNDM architectures with up to 5 hidden layers and 512 neurons per layer, each with ReLU activation function. The dimensionality and architecture information is provided in Table 1.

The state space of the Pendulum consists of the pole angle $\theta$ and angular velocity $\dot{\theta}$, with the safe set defined as $[-\pi/15, \pi/15] \times [-1, 1]$, and the initial set as $\theta_0 \in [-\pi/36, \pi/36]$. For the Cartpole, the state space consists of the cart position $x$, velocity $\dot{x}$, pole angle $\theta$, and angular velocity $\dot{\theta}$. The safe sets for cart position are defined as $x \in [-1, 1]$ and $\theta \in [-\pi/15, \pi/15]$, with the initial set $\theta_0 \in [-\pi/36, \pi/36]$. For the 4D Husky, the states are position $x$ and $y$, orientation $\theta$, and linear velocity $v$. The task of the controller is to keep the robot within a lane, and hence the safe set is $y \in [-1, 1]$, and the initial set is defined as any position $(x, y)$ that is within a radius of 0.1 from the

Table 1: Benchmark results for barrier certificate and control synthesis on various NNDMs. Threshold $\delta_s = 0.95$ was used for the minimally-invasive control synthesis. Architecture $h \times [r]$ denotes a NNDM with $h$ hidden layers, each with $r$ neurons. $n$ is the dimensionality of the system.

| Model | $n$ | $h \times [r]$ | $|Q|$ | Verification | | | | | | Control | | | | | |
| | | | | Interval Bounds | | | Linear Bounds | | | Interval Bounds | | | Linear Bounds | | |
| | | | | $\beta$ | $P_s$ | Time (min) | $\beta$ | $P_s$ | Time (min) | $\beta$ | $P_s$ | Time (min) | $\beta$ | $P_s$ | Time (min) |
|---|---|---|---|---|---|---|---|---|---|---|---|---|---|---|---|
| Pendulum | 2 | 1 x [64] | 120 | 0.531 | 0.468 | 0.06 | 0.005 | 0.995 | 0.09 | $10^{-6}$ | 0.999 | 0.04 | - | - | - |
| | | | 240 | 0.430 | 0.569 | 0.14 | 0.005 | 0.995 | 0.14 | $10^{-6}$ | 0.998 | 0.11 | - | - | - |
| | | | 480 | 0.146 | 0.854 | 0.36 | 0.005 | 0.995 | 0.43 | $10^{-6}$ | 0.999 | 0.32 | - | - | - |
| | | 2 x [64] | 120 | 0.555 | 0.444 | 0.05 | 0.196 | 0.782 | 0.06 | $10^{-6}$ | 0.999 | 0.04 | $10^{-6}$ | 0.999 | 0.05 |
| | | | 240 | 0.440 | 0.556 | 0.14 | 0.157 | 0.841 | 0.15 | $10^{-6}$ | 0.997 | 0.11 | $10^{-6}$ | 0.998 | 0.13 |
| | | | 480 | 0.196 | 0.802 | 0.46 | 0.079 | 0.919 | 0.48 | $10^{-6}$ | 0.998 | 0.32 | $10^{-6}$ | 0.999 | 0.38 |
| | | 3 x [64] | 120 | 0.674 | 0.320 | 0.05 | 0.388 | 0.597 | 0.06 | $10^{-6}$ | 0.994 | 0.06 | $10^{-6}$ | 0.985 | 0.05 |
| | | | 240 | 0.534 | 0.461 | 0.14 | 0.293 | 0.703 | 0.17 | $10^{-6}$ | 0.995 | 0.11 | $10^{-6}$ | 0.996 | 0.12 |
| | | | 480 | 0.356 | 0.636 | 0.41 | 0.211 | 0.788 | 0.51 | $10^{-6}$ | 0.999 | 0.39 | $10^{-6}$ | 0.993 | 0.47 |
| | | 5 x [64] | 480 | 0.868 | 0.030 | 0.40 | 0.827 | 0.124 | 0.44 | $10^{-6}$ | 0.907 | 0.37 | $10^{-6}$ | 0.951 | 0.34 |
| | | | 960 | 0.722 | 0.229 | 1.55 | 0.692 | 0.259 | 1.67 | $10^{-6}$ | 0.909 | 1.42 | $10^{-6}$ | 0.951 | 1.61 |
| | | | 1920 | 0.503 | 0.448 | 5.82 | 0.458 | 0.492 | 6.28 | $10^{-6}$ | 0.911 | 5.33 | $10^{-6}$ | 0.951 | 5.39 |
| Cartpole | 4 | 1 x [128] | 960 | 1.00 | 0.00 | 17.41 | 0.625 | 0.375 | 17.65 | $10^{-6}$ | 0.877 | 12.43 | $10^{-6}$ | 0.998 | 15.53 |
| | | | 1920 | 1.00 | 0.00 | 57.83 | 0.417 | 0.574 | 65.15 | $10^{-6}$ | 0.900 | 31.31 | $10^{-6}$ | 0.993 | 35.09 |
| | | | 3840 | 0.796 | 0.193 | 216.92 | 0.385 | 0.599 | 224.70 | $10^{-6}$ | 0.979 | 158.51 | $10^{-6}$ | 0.985 | 164.41 |
| | | 2 x [128] | 960 | 1.00 | 0.00 | 17.80 | 0.779 | 0.172 | 18.19 | $10^{-6}$ | 0.816 | 11.98 | $10^{-6}$ | 0.951 | 14.02 |
| | | | 1920 | 1.00 | 0.00 | 35.44 | 0.773 | 0.178 | 34.71 | $10^{-6}$ | 0.849 | 31.98 | $10^{-6}$ | 0.997 | 36.44 |
| | | | 3840 | 0.878 | 0.101 | 228.78 | 0.666 | 0.331 | 225.42 | $10^{-6}$ | 0.980 | 148.49 | $10^{-6}$ | 0.997 | 160.46 |
| Husky | 4 | 1 x [256] | 900 | 0.672 | 0.200 | 14.23 | 0.691 | 0.259 | 16.80 | $10^{-6}$ | 0.872 | 11.22 | $10^{-6}$ | 0.951 | 11.71 |
| | | | 1800 | 0.667 | 0.211 | 52.70 | 0.673 | 0.278 | 58.76 | $10^{-6}$ | 0.878 | 33.27 | $10^{-6}$ | 0.951 | 35.10 |
| | | | 2250 | 0.645 | 0.288 | 69.12 | 0.368 | 0.494 | 81.32 | $10^{-6}$ | 0.933 | 42.01 | $10^{-6}$ | 0.951 | 46.65 |
| | | | 4800 | 0.622 | 0.331 | 300.10 | 0.347 | 0.539 | 345.84 | $10^{-6}$ | 0.951 | 298.71 | $10^{-6}$ | 0.953 | 302.31 |
| | | 2 x [256] | 1800 | 1.00 | 0.00 | 64.61 | 0.544 | 0.010 | 74.62 | $10^{-6}$ | 0.749 | 30.44 | $10^{-6}$ | 0.800 | 34.99 |
| | | | 2250 | 1.00 | 0.00 | 69.76 | 0.502 | 0.222 | 74.80 | $10^{-6}$ | 0.847 | 49.01 | $10^{-6}$ | 0.951 | 52.02 |
| | | | 4800 | 0.845 | 0.062 | 364.45 | 0.415 | 0.384 | 400.88 | $10^{-6}$ | 0.908 | 295.75 | $10^{-6}$ | 0.951 | 274.94 |
| Husky | 5 | 1 x [512] | 432 | 1.00 | 0.00 | 12.43 | 1.00 | 0.00 | 11.82 | $10^{-1}$ | 0.750 | 7.21 | $10^{-6}$ | 0.949 | 9.03 |
| | | | 1080 | 1.00 | 0.00 | 51.59 | 1.00 | 0.00 | 57.65 | $10^{-3}$ | 0.829 | 49.38 | $10^{-6}$ | 0.950 | 51.76 |
| | | | 1728 | 1.00 | 0.00 | 168.39 | 1.00 | 0.00 | 171.35 | $10^{-3}$ | 0.951 | 153.59 | $10^{-6}$ | 0.951 | 161.04 |
| Acrobot | 6 | 1 x [512] | 144 | 1.00 | 0.00 | 11.10 | 1.00 | 0.00 | 12.02 | $10^{-6}$ | 0.951 | 4.44 | $10^{-6}$ | 0.951 | 4.93 |
| | | | 288 | 1.00 | 0.00 | 25.01 | 0.863 | 0.088 | 25.62 | $10^{-6}$ | 0.951 | 11.55 | $10^{-6}$ | 0.951 | 11.78 |

origin. For the 5D Husky, the additional state over the 4D model is the angular velocity $\omega$; the task, the safe set, and the initial set remain the same. Finally, for the Acrobot, the state space consists of the cosines and sines of $\theta_1$ and $\theta_2$, and angular velocities $\dot{\theta}_1$ and $\dot{\theta}_2$. Here $\theta_1$ and $\theta_2$ are the pole angle of the first link and the angle of the second link relative to the first link, respectively. The task is for the tip of the second link to reach a height of at least $y = 1$, with a safety constraint to never reach beyond $y = 1.2$. Thus, the safe set is defined as $\sin(\theta_1) \in [-0.6, 0.6]$ and $\sin(\theta_2) \in [-0.6, 0.6]$, and the initial set is any point within a radius of 0.1 around the origin in the first 4 dimensions.

**Implementation and Experimental Setup:** We implemented our algorithms in Python and Julia (code available in [43]). For collecting data and training, we used the TensorFlow framework [44]. To compute the linear approximations of the NNDMs, we utilized $\alpha$-CROWN [36]. For the optimization, we used Julia's *SumOfSquares.jl* package [45, 46]. The optimizations are all performed single-threaded on a computer with 3.9 GHz 8-Core CPU and 64 GB of memory.

**Benchmarks:** To the best of our knowledge, there is no other work on safety verification of NNDMs, against which we can compare. Hence as a baseline, we compare our approach based on linear under and over approximation of the NNDM against interval bounding approaches cf. Remark 5. The interval bounding approach results in a relaxed optimization problem at the cost of more conservative safety probabilities. The results are shown in Table 1. The computation times indicate the time to synthesize the barrier and the controller using Theorems 1 and 2, respectively. The barrier degree is 4 for all the case studies. Below, we provide a brief discussion on the results. For more in-depth discussions and details on the discretization and hyperparameters, see Appendix A.

**Verification:** As expected, the safety probability always increases with $|Q|$ (finer partitions), since the linear and interval bounds for $f^w$ become more accurate. This behavior is encountered in all the systems and architectures, regardless of the number of hidden layers or neurons per layer. However, it is also observed that this trend of finer discretizations and increased safety probability consistently comes at the cost of an increased computation time. This is due to the fact that the number of partitions directly dictate the number of constraints in the optimization problems. Note that the probability of safety using interval bounds is strictly worse than linear approximations for all the experiments. Using either linear or interval bounds, we are able to compute non-trivial probability of safety for all the case-studies and observe the discretization vs accuracy trend. Take for instance, the 1-layer Cartpole model, where an increase in $|Q|$ from 960 to 3840 causes an increase in $P_s$ from

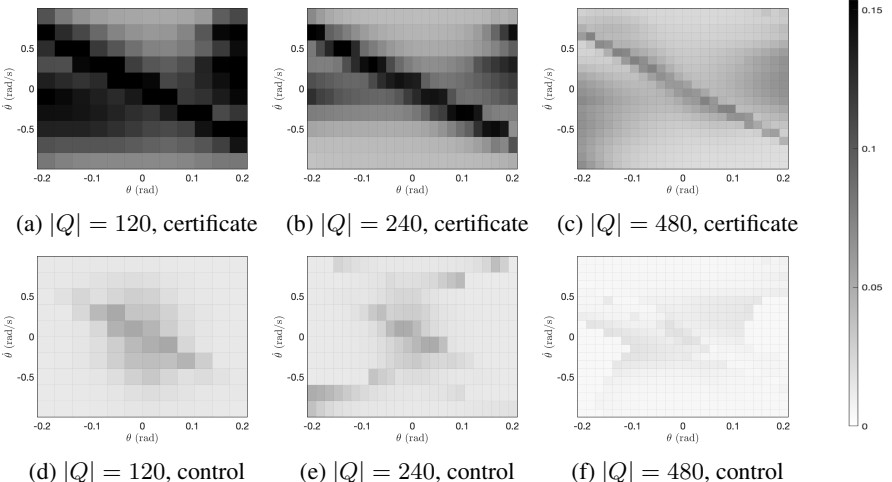

(a) $|Q| = 120$, certificate    (b) $|Q| = 240$, certificate    (c) $|Q| = 480$, certificate

(d) $|Q| = 120$, control    (e) $|Q| = 240$, control    (f) $|Q| = 480$, control

Figure 1: Comparison of $\beta_q$ values for the Pendulum NNDM with 2 layers and 64 neurons per layer before and after applying the controller for various $|Q|$. Shade of gray correspond to the values of $\beta_q$.

0.375 to 0.599, at the expense of an increased computation time of more than 12 times. This behavior is also true for the 4D Husky model, where increasing $|Q|$ from 1800 to 4800 for the 2-layer model, increases $P_s$ from 0.010 to 0.384 at the price of a more than 5-fold increase in the computation time.

**Control Synthesis:** When the certification probability is below $\delta_s = 0.95$, the control strategy is utilized to increase the probability of safety. Using interval bounds, controllers had to be synthesized for all case-studies, which illustrates the looseness of the bounds. In fact, in several case studies, the controller was not able to meet the desired threshold. For the linear bounds, observe that for all except two NNDMs, whose $P_s < \delta_s$, the controller synthesis algorithm was able to produce $P_s \geq \delta_s$, showing the effectiveness of our control method. The first exception is for the 4D Husky model with 2-hidden layers and $|Q| = 1800$, where the controller increases $P_s$ from 0.010 to 0.800 (73-fold increase) yet cannot meet the 0.95 threshold. Similarly, for the 1-layer 5D Husky model with $|Q| = 432$, the controller increases $P_s$ from 0.00 to 0.949, but the 0.95 threshold cannot be reached. Nonetheless, for all the linear bound cases, the controller reduces $\beta$ to $10^{-6}$.

Figure 1 illustrates the minimally-invasive aspect of the controllers. It shows the $\beta_q$ values for the Pendulum NNDM with 2 layers before and after the application of the controllers with three different sizes of $|Q|$. For $|Q| = 120$, all the regions require controllers, but for $|Q| = 240$ and 480, respectively, only 69.2% and 23.75% of the regions require controllers; hence, the minimally-invasive controller focuses specifically on those regions and does not affect those with already-small $\beta_q$.

# 7 Conclusion

In this work, we introduced a methodology for providing safety guarantees for NNDMs using stochastic barrier functions. We showed that the problem of finding a stochastic barrier function for a NNDM can be relaxed to an SOS optimization problem. Furthermore, we derived a novel framework for synthesis of an affine controller with the goal of maximizing the safety probability in a minimally-invasive manner. Experiments showcase that, along with a minimally-invasive controller, we can guarantee safety for various standard reinforcement learning problems.

A bottleneck to our approach is the discretization step, which is required to obtain piece-wise linear under- and over-approximations of the neural network, and can be expensive for higher dimensional spaces. Future directions should focus on mitigating this curse of dimensionality.

# Acknowledgments and Disclosure of Funding

This work was supported in part by the NSF grant 2039062 and NASA COLDTech Program under grant #80NSSC21K1031.

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
