# OpenReview forum: "Safety Guarantees for Neural Network Dynamic Systems via Stochastic Barrier Functions"
_NeurIPS.cc/2022/Conference — NeurIPS 2022 Accept_

### Official Review · Reviewer_D3ig · 2022-07-06

**Rating:** 7
**Confidence:** 2
**Soundness:** 3 good
**Presentation:** 3 good
**Contribution:** 3 good

**Summary:**

This work provides probabilistic safety guarantees for dynamical systems. Noticing the safety issues in neural network based dynamical systems, this paper utilizes stochastic barrier function generated by SOS optimization and propose a safety probabilistic certificate for discrete step system. In the control synthesis problem, the paper also provides a lower bound of safety probability by local convex relaxation.

**Questions:**

1. I’m a bit concerned on the comparison metrics in the experiment, see my detailed question 1,2.

2. The computational time could be an issue for high dimensional system and finer partitions, but this is not a deal-breaker since the target is certification.

**Limitations:**

1. Algorithm 1 has safe set as input, how to determine unsafe/safe region for each application? If that requires additional prior knowledge, then it might be unfair to compare the safety probability to safe region between synthesis controller and baseline model from imitation learning, since the former one already “knows” where the boundary is.

2. As for the verification comparison itself, I’m concerned why use the model from imitation learning as baseline instead of the expert controller. If the imitation learned model only saw the trajectory generated from an expert controller during training, then it’s reasonable it cannot handle corner cases. In line 311, it seems like your synthesis controller has been exposed to new data, then this is again not fair comparison. Feel free to correct me if I misunderstood your approach.

3. In the verification results, I’m curious why larger architecture leads to worse Ps?




**Strengths And Weaknesses:**

1. Introducing the stochastic function into dynamical system learning is novel.

2. The narrative structure is clear.

3. It’s interesting to see many tricks in this paper to reformulate the problems.

---

> ### Author Response · Authors · 2022-08-02
> **Clarification for high dimensional systems, comparison metrics, and approximation of large NNs**
>
> Thank you for your insightful comments: we appreciate your feedback.
>
> > The computational time could be an issue for high dimensional system and finer partitions, but this is not a deal-breaker since the target is certification…
>
> We agree with the reviewer that scalability to high dimensional systems could be an issue. However, we emphasize that (i) the goal of this paper is to lay down the theoretical foundation for using barrier functions to certify NNDM, and the experiments validate the developed theory, and (ii) there are various strategies that we can are take to improve scalability:
>
> - a) we can rely on adaptive partitioning schemes that partition regions with high $\beta$ values and merge regions with low $\beta$ values. Such branch-and-bound schemes are known to often produce drastic improvement in performance [A6]
>
> - b) we can devise decentralized approaches that allow for parallelization of the optimization problem to construct the barrier function independently for different regions.
>
> [A6] Royo, Vicenc Rubies, et al. "Fast neural network verification via shadow prices", 2019
>
> > Q1: Algorithm 1 has safe set as input, how to determine unsafe/safe region for each application? If that requires additional prior knowledge, then it might be unfair to compare the safety probability to safe region between synthesis controller and baseline model from imitation learning, since the former one already “knows” where the boundary is…
>
> Motivated by robotic applications, where the safety constraints of the system are typically known, in this work as formulated in Problem 1, we assume that the initial and safe sets as well as the NNDM are given. To obtain the NNDM for our experiments, we use imitation learning.
>
> We emphasize that the purpose of the benchmarks is to show the effectiveness of the barrier controller. Our goal is not to show that the underlying closed-loop system (NNDM) has a bad controller.  To ensure this point is clear and the comparisons are fair, we have added a new baseline comparison that focuses on the approach rather than the NNDM as suggested by Reviewer JTrX.  Please see our response to Q1 of Reviewer JTrX and the revised paper.
>
> > Q2: As for the verification comparison itself, I’m concerned why use the model from imitation learning as baseline instead of the expert controller. If the imitation learned model only saw the trajectory generated from an expert controller during training, then it’s reasonable it cannot handle corner cases…
>
> As per the problem formulation, we are given a NNDM that represents the dynamics and controller as a whole. In our experiments, we use imitation learning to obtain the NNDM, but our method is not limited to that. In general, the NNDM can be obtained using various methods (e.g., RL method in [9]). Our framework assumes a NNDM is given.
>
> Our goal is to show that we can preserve the behavior of the NNDM while intervening only in the unsafe regions in a minimally invasive fashion to achieve the desired safety threshold.
>
> For instance, for the husky model, we do not have access to the true dynamical model. For this reason, the NNDM was trained using data collected from a physics simulator. Hence, we compare the behavior of the learned system with and without the barrier controller.
>
> > Q3: In line 311, it seems like your synthesis controller has been exposed to new data, then this is again not a fair comparison.
> Feel free to correct me if I misunderstood your approach…
>
> For the husky system, we train a NN that represents the learned dynamics and a NN controller that emulates a model-predictive controller used to train the NN dynamics model. The toolbox we use to compute the linear approximation can not handle the two NNs conjoined as one due to theoretical limitations. Thus, we train a NNDM by collecting input-output data from the previous learned dynamics and controller NN models, and compute linear approximation for a single NN.
>
> > Q4: In the verification results, I’m curious why larger architecture leads to worse Ps?…
>
> This is a good question. Larger architectures produce looser linear over- and under-approximations for the NN [33]. This error leads to a conservative barrier and loose lower bound for the probability of safety.  To reduce this error, smaller discretization of the safe set is required, which leads to higher computation cost.
>
> For the computation of the linear approximation on the NN, we use the method in [32,33]. It first back-propagates the linear bounds by computing linear relaxations [33, Equation 7, matrix  D] of the output of each neuron from the output layer, through the hidden layers, till it reaches the input layer of the NN. The tightness of these bounds depends on the depth of the architecture, the width of each layer, the magnitude of weights in the NN, and the size of the input domain (width of the hyper-rectangle). This is explained in Lemma 2.1 of [32].

---

### Official Review · Reviewer_crLS · 2022-07-09

**Rating:** 3
**Confidence:** 4
**Soundness:** 2 fair
**Presentation:** 2 fair
**Contribution:** 2 fair

**Summary:**

This paper considers the safety guarantees of the neural network dynamic models (NNDMs). Specifically, the authors use stochastic barrier certificate to verify if a given NNDM satisfy the safety in a probabilistic way, which provides a lower bound on the system's safety probability. If the safety probability is below the desired threshold, then the authors propose to directly modify the output of the NNDM that robustly minimize the unsafety probability in a minimally-invasive manner. The authors applied the proposed method on three different control problems.

**Questions:**

1. In line 15, the authors claim they introduce a method of generating controls for the system. What is the control of the system? The authors just directly modify the system state in the form of equation (3). This is not the real control of the system.

2. It is hard to justify the contribution of this paper since the authors just modify the output of a NNDM (equation (3)) to satisfy the safety probability.

3. In line 112, the authors claim that ``Computation of this probability is particularly challenging because system (1) is stochastic and NN is generally non-convex'' ? I do not think this is a reasonable motivation.

4. In line 118, I do not think u_k is the ``real'' control action of the system. It is just a variable to change the output of a NNDM.

5. In equation (4b), what is $\eta$? Could it be negative?

6. The right hand side of (5) is very likely to be negative. The authors should justify this.

7. In Proposation 1, what is $\lambda_0(x)$? Moreover, in Corollary 1, the conclusion is equivalent to state that $\lambda_0(x)$ is a SOS polynomial following the last equation of Proposation 1. This is really confusing.

8. In line 181, can we always partition $X_s$ to a finite set of regions, in which each region is a basic semi-algebraic set?

9. In theorem 1, what are the domains of $\beta$ and $\eta$? Moreover, in theorem 2, is this optimization always feasible? For example, what if $\theta$ is negative? The authors should clearly define the domain of a variable before using it.

10. In experiments, there are no comparisons with other approaches that consider safety guarantees. Therefore, it is hard to justify the contribution of this work. Moreover, the authors did not explain why there is an exception in Husky, in which case the safety probability is 0.800 (cannot meet the 0.95 threshold).


**Ethics Review Area:**

["I don’t know"]

**Limitations:**

The authors just modify the output of a NNDM to satisfy a given safety probability. This approach does not generate a ``real'' control for the system. Therefore, it does not apply to dynamical systems.

**Strengths And Weaknesses:**

Strengths:

This paper combines stochastic barrier certificate and convex relaxation of a neural network in a good way.

Weaknesses:

The contibutions of this paper are not clear. There are some related works, and the authors have not compared their method with other approaches, especially in the experiments.  There is one important notation that is misleading. When the safety probability is below the desired threshold, the authors just modify the output of the NNDM, instead of generating controls for the system as claimed by the authors. The ``control'' of a system should not be considered in the form of equation (3). The equation (3) just directly modify the output of the NNDM. For example, if the state x denotes the system's position that is controlled by the system's acceleration, the authors just modify the position to satisfy the safety probability instead of changing the acceleration. In order to model the real control of a system, the authors should consider the control in the $f^w$. In other words, the control is an input to the NNDM.

---

> ### Author Response · Authors · 2022-08-02
> **Clarifications on the Controller and Eq. (3)**
>
> Thank you for your insightful comments: we appreciate your feedback.
>
> > Q1: The authors just directly modify the system state in the form of equation (3). This is not the real control of the system…
>
> > Q2: hard to justify the contribution of this paper since they just modify the output of a NNDM (equation 3)…
>
> > Q4: In line 118, I do not think u_k is the ``real'' control action of the system. Just a variable to change the output of a NNDM…
>
> We would like to stress that Equation (3) does not necessarily imply that control input $u\in\mathbb{R}^c$ is applied to every component of state $x\in\mathbb{R}^n$, i.e., ($c$ and $n$ are not necessarily equal). In that equation, vector $u$ is pre-multiplied by matrix $g\in\mathbb{R}^{n \times c}$, which is system dependent and maps controllers to additive terms for only the state components that are affected by the controller.
>
> In fact, none of the models in our experiments allows the control of all of the state variables. The controls (actions) are exactly the ones indicated in the OpenAI Gym.
>
> - Pendulum: angular acceleration
> - Cartpole: linear and angular accelerations
> - Husky: linear acceleration and angular velocity
>
> as detailed in Appendix B.1 in the supplementary material.
>
> We further note that Equation (3) represents control-affine systems, which constitute a large class of control systems, including most robotic systems. In Problem 2, we implicitly assume the system is control affine.  We have made this assumption explicit in Problem 2 (revised version).
>
> > Q3: Line 112, `Computation of this probability is particularly challenging because system (1) is stochastic and NN is generally non-convex'' ? I do not think this is a reasonable motivation…
>
> The purpose of that sentence is to describe why the problem is difficult; not necessarily to motivate it. To compute the probability of safety, accurate forward propagation of the state uncertainty in time via the dynamics is needed. Our dynamics are given by a NN (nonlinear, non-convex), making accurate computation of $P_s$ very difficult.
>
> > Q5: In equation (4b), what is η? Could it be negative?…
>
> $\eta$ is an upper bound on the barrier function in the initial set $X_0\subseteq X$.  It cannot be negative because $B(x)\geq0$ for all $x\in X$, as indicated in Condition (4a).
>
> > Q6: RHS of (5) is very likely to be negative. Should justify…
>
> It is possible for the RHS of (5) to be negative, which would be a trivial lower bound for $P_s$.  Perhaps, a more accurate presentation would be $P_s\geq\max\set{0,RHS \ of\(5)}$, but Equation (5) is the standard form in the literature on stochastic barrier functions [8],[A2-A5].
>
> [A2] Papachristodoulou et al. “SOSTOOLS - Sum of Squares Optimization Toolbox for MATLAB”, 2021
>
> [A3] Parrilo, “Structured Semidefinite Programs and Semialgebraic Geometry Methods in Robustness and Optimization”, 2000
>
> [A4] Dutreix et al. “Interval-valued Markov Chain Abstraction of Stochastic Systems using Barrier Functions”, 2020
>
> [A5] Jagtap et al. “Formal synthesis of stochastic systems via control barrier certificates”, 2019
>
> > Q7: In Prop. 1, what is λ0(x)? In Corollary 1, the conclusion is equivalent to state that λ0(x) is a SOS polynomial following the last equation of Prop 1…
>
> Thank you for pointing out this confusion. In Prop. 1, $\lambda_0(x)$ is an SOS polynomial. To reduce confusion, we explicitly define $\lambda_0(x)\in\Lambda$ in Prop. 1.
>
> > Q8: In line 181, can we always partition Xs to a finite set of  basic semi-algebraic-set regions?…
>
> In Problem 1, $X_s$ is assumed to be a basic closed semi-algebraic set, which can always be partitioned to a finite set of regions, e.g., by using half spaces.
>
> > Q9: Theorem 1, what are the domains of β and η?…
>
> As introduced in Sec. 3.1, $\beta,\eta\in[0,1]$ (clarified in the revised version).
>
> > Q10: Theorem 2, is this optimization always feasible? …
>
> Variable $\theta\in\mathbb{R}_{\geq 0}$ (clarified in the revised version). Then, the LP in Theorem 2 is always feasible because the feasible set is non-empty for $\underline{f}_q(x)\leq\bar{f}_q(x)$, which is assumed in Equation (6).
>
> > Q11: In experiments, no comparisons…why is there an exception in Husky, in which case the safety probability is 0.800 (cannot meet the 0.95 threshold)…
>
> To the best of our knowledge, there is no other work on safety verification of NNDMs, which makes it difficult to compare our framework with other approaches. Nevertheless, we have added a comparison benchmark in the revised version (please see our response to Q1 of Reviewer JTrX and the paper).
>
> Re: Husky, this is a good observation. The initial probability of safety is just 0.011. The controller is able to increase this safety probability to 0.800 but still cannot meet the threshold of 0.95. That is because the controller is bounded and cannot overcome the error introduced by the linear bounds of the NNDM. When the partition is refined, this error reduces and the controller is able to successfully meet the threshold.

---

> > ### Comment · Reviewer_crLS · 2022-08-08
> > **Final comments**
> >
> > I would like to thank the authors for addressing my questions.  The questions here are for improving the clarifications. However, my main concerns are the contributions and limitations:
> >
> > 1.  The main confusion of the dynamics (3) is due to the fact that $g$ is not a neural network, not to mention that $g$ is totally independent of the state $x_k$. I think it is reasonable to consider affine control systems. However, since $g$ is just a constant matrix here and the dynamics (3) are in discretization form, we could view dynamics (3) as a way of directly modifying the output of the neural network. Therefore, I think the considered problems of this paper is very limited.
> >
> > 2. Barrier certificate has already been widely used in optimization and control for safety-critical systems, as well as in neural networks (e.g., Gruenbacher, et.al. On the verification of neural odes with stochastic guarantees, AAAI2021; Lechner, et.al., Infinite time horizon safety of Bayesian neural networks, NeurIPS 2021; Dawson, et.al., Safe Control with Learned Certificates: A Survey of Neural Lyapunov, Barrier, and Contraction methods). Safe verification for NNDMs is just one of these problems. Therefore, I think the contribution of this work is limited.
> >
> > 3. Regarding the question 11, I do not see how the control bound contributes a limitation here. This should be better justified and clarified in terms of the feasibility guarantees.

---

> > > ### Author Response · Authors · 2022-08-09
> > > **Generality of the work**
> > >
> > > We thank the reviewer for their reply.
> > >
> > > > 1. The main confusion of the dynamics (3) is due to the fact that g is not a neural network, not to mention that g is totally independent of the state . I think it is reasonable to consider affine control systems. However, since g is just a constant matrix here and the dynamics (3) are in discretization form, we could view dynamics (3) as a way of directly modifying the output of the neural network. Therefore, I think the considered problems of this paper is very limited.
> > >
> > > We apologize for not having clarified this point before: for the sake of a simpler presentation, in Eq. (3), $g$ is taken to be a constant; however, our approach is not limited to that and can also support the case where $g$ is a continuous function of $x$ or a neural network. In fact, in the case where $g$ is a neural network, we can simply use the interval bound propagation approach introduced in Section 4.1 to find for each partition $q$, \
> > > $\underline{g_q} \leq \min_{x \in q} g(x)$ and $\overline{g_q} \geq \max_{x\in q} g(x)$ \
> > > and then include them in the constraints of Theorem 2. Note that in this more general case a product of variables may appear in Eq. (10d) and (10e). However, the resulting problem can still be reduced to LP by introducing additional variables and constraints as described in Section 7.7 in [R1]. In order to clarify this point and the modeling flexibility of our framework, we will add a remark to the paper.
> > >
> > > [R1] Aimms, B. V. "Aimms modeling guide—integer programming tricks." Pinedo, Michael. Scheduling: Theory, Algorithms, and Systems; AIMMS BV (2016).
> > >
> > > >  2. Barrier certificate has already been widely used in optimization and control for safety-critical systems, as well as in neural networks (e.g., Gruenbacher, et.al. On the verification of neural odes with stochastic guarantees, AAAI2021; Lechner, et.al., Infinite time horizon safety of Bayesian neural networks, NeurIPS 2021; Dawson, et.al., Safe Control with Learned Certificates: A Survey of Neural Lyapunov, Barrier, and Contraction methods). Safe verification for NNDMs is just one of these problems. Therefore, I think the contribution of this work is limited.
> > >
> > > We thank the reviewer for the references. However, while it is true that barrier functions have already been used in safety critical applications, we would like to stress that our work is *substantially* different than those referenced by the reviewer. In particular, the key differences are:
> > >
> > > 1. None of those works consider neural network dynamic models or noisy dynamics. Most of those works only consider neural network controllers and polynomial dynamics. The ability to handle non-convex and uncertain dynamics via SOS optimization is a major contribution of our work.
> > > 2. Referenced papers are limited to two or at most three dimensional systems. As shown in the reply to Reviewer JTrX, our approach can already scale to 6 dimensional systems and has the potential to scale to even larger systems.
> > > 3. Our control synthesis approach that is based on linear programming is another major contribution of the work, which can also be employed even for control systems with analytical dynamics models.
> > >
> > > As a consequence, we believe that the contributions of our work are not limited.  To make sure these points are clarified in the paper, we will add a discussion in the related work section on the differences between this work and those papers.
> > >
> > > >  3. Regarding the question 11, I do not see how the control bound contributes a limitation here. This should be better justified and clarified in terms of the feasibility guarantees.
> > >
> > > We acknowledge that our previous response wasn’t very clear.  As shown in Alg. 1, we attempt to obtain a satisfactory probability of safety through an iterative process of synthesizing a barrier with an increasing bound on $\eta$ and generating controllers to reduce $\beta_q$. In that particular experiment, we could not generate a valid barrier until the bound on $\eta$ reached 0.2 despite the best control efforts to reduce $\beta_q$. With $\eta \leq 0.2$, a valid barrier was synthesized whose $\beta_q$ could be derived to $10^{-6}$, giving the best safety probability of 0.8.  We will expand on this in the paper.

---

### Official Review · Reviewer_JTrX · 2022-07-11

**Rating:** 6
**Confidence:** 4
**Soundness:** 4 excellent
**Presentation:** 3 good
**Contribution:** 3 good

**Summary:**

The paper introduces a safety-guaranteeing controller distillation method for neural network dynamic models (NNDM). The approach is based on the stochastic barrier functions. The introduced algorithm works in two phases. In the first phase, a convex approximation results are applied to obtain piecewise linear bounds for the NNDM on the domain discretized into square regions. In the second step, a synthesis of a minimally invasive (according to the provided definition) controller is carried out using the linear programming approach.

**Questions:**

* There is no baseline presented for the experimental section, which makes the presented results incomparable, and  hard to evaluate the impact of the algorithm being introduced;
* It would be very nice if authors use at least one example, in which an analytical model is not available or is more complicated than a nonlinear ODE/hybrid system, appearing in the classical applications of barrier function certificates to closed-form models, maybe one of the  MuJoCo standard environments should do a good job here;
* the experimental setting is not clearly introduced in the main part, e.g. the safety certification sets used for the barrier function computation are not provided;
* l. 309, the Husky models used for the experimental evaluation are not standard, haven't seen them being used in RL literature, please explain and provide an appropriate reference;
* regarding the implementation, what is the rationale in having part of the code (i.e. NeuralNetControlBarrier) written in Julia, as opposed to having the code fully written in Python ?

## Minor remarks
* l. 261, elaborate and provide a reference for 'called Jensen’s gap, is a non-negative polynomial of x, u, and moments E[vd]';
* l. 217 the difference between $L_q(x)$ and $L_{q,x}(x) $ is not explained;

**Limitations:**

The limitations related especially to the scalability issue have been appropriately addressed in the paper.

**Strengths And Weaknesses:**

The paper introduces a novel approach for a problem of large importance in safe machine learning, i.e. synthesis of safe controllers for NNDM's. For instance, NNDM's are widely encountered in model-based reinforcement learning approaches, and a synthesis of a safe controller given NNDM has a large potential for practical applications.

The paper aims at bringing the tool of barrier functions previously widely studied in control theory approaches for closed-form models (restricting practical applications in ML) into the world of NN-based models. I have not verified the claim that this is in fact the first paper demonstrating such an approach, but if yes then the paper would constitute a major result.

The approach is reasonable and is clearly presented. The formal presentation is supported by theoretical results, i.e. lemmas and theorems. The layout makes it easy to follow and understand the algorithm construction. The overall approach seems to be sound. The formal presentation is supported by experimental case studies (Sec. 6).

I am happy with the formal presentation but have some objections to the experimental evaluation. My main critique, see also questions below, is lack of proper presentation of the setup in the main paper. I would rethink if all of the plots in Fig. 1 are crucial for presentation, and perhaps replace some of them with more illuminating text. The examples, confirm that the approach works in a desirable way (large safety certificate probability/ small $\varepsilon$ of the minimally invasive controller) , however, those results are not compared to a single baseline, which makes them harder to appreciate. The third is the scalability issue, it is actually noted by the authors that the bottleneck is the discretization step. In fact, it is hard to think how such an approach would scale up to larger systems, given that the phase space needs to be discretized into square-like subregions, resulting in an exponential explosion of the number of subregions. Without overcoming this issue the approach may not have a significant impact in reliable ML community, and I would like the authors to at least provide a convincing argument for that there exists a path towards relaxing the restrictive discretization step.

Weighting positives and negatives, I think the paper is a valuable contribution to the reliable machine learning field of research, and I am leaning toward accepting it.
 I am voting for weak accept at present, but I am open to increasing my score in case the authors satisfactorily address my remarks for the experimental section of the work, i.e., will provide some baseline method evaluation, will improve the presentation of the experimental results in the main part and will discuss how the restrictive discretization step could be circumvented..

---

> ### Author Response · Authors · 2022-08-02
> **Clarification of experimental setup and added benchmarks**
>
> Thank you for your insightful comments: we appreciate your feedback.
>
> > Q1: There is no baseline presented for the experimental section…
>
> To the best of our knowledge, there is no other work on safety verification of NNDMs, which makes it difficult to compare our framework with other approaches. As a baseline, however, we can compare our approach based on linear under and over approximation of the NNDM against interval (global) bounding approaches. In particular, interval bounds only bound the value of a NNDM in a partition with (an over-approximation of) the supremum and infimum of the value of the neural network in that partition using Corollary 3.3 in [18]. This results in a simpler optimization problem compared to that in Theorem 1, but at the cost of more conservative results.
>
> These benchmarks are added to the revised version. As shown in Table 1, our approach in all the benchmarks substantially returns higher values of safety probability at the cost of a slight increase in the computation time.
>
> > Q2: It would be very nice if authors use at least one example, in which an analytical model is not available…
>
> We would like to stress that, while for the Pendulum and Cartpole NNDMs, the data were collected from known analytical models (from OpenAI gym) under a given expert controller (OpenAI Gym leaderboard), for the Husky there is not an analytical model available. In particular, for the Husky all NNDMs have been trained in PyBullet [42], which is a physics simulator similar to MuJoCo. The lack of an analytical model is indeed one of the main reasons we decided to include the Husky as one of our benchmarks. Please see the response to Q4 for a more detailed description of the Husky benchmark.
>
> > Q3: the experimental setting is not clearly introduced in the main part…
>
> We have clarifIed this information in the revised version. In particular, the state space of the pendulum is the pole angle and angular velocity, and the safe set is [-$\pi$/15,$\pi$/15] and [-1,1], respectively, and the initial set of the pole angle is  [-$\pi/36$,$\pi/36$].
>
> For the cartpole, the state space is the cart position and velocity, and pole angle and angular velocity. The safe set is [-1,1] for the cart position and [-$\pi$/15,$\pi$/15] for the pole angle, and the initial set is [-$\pi/36$,$\pi/36$] for the pole angle.
>
> For husky, the states we observe are position $x$ and $y$, orientation $\theta$, and linear velocity $v$. The task is to stay within a lane and hence $y$ should be between [-1,1], and the initial set is the set of all states whose $x$ and $y$ components are within the radius of 0.1 from the origin.
>
> > Q4: l. 309, the Husky models used for the experimental evaluation are not standard…
>
> The Husky is a 4-dimensional model of a robot designed by Clearpath Robotics, which is trained using Pybullet [42], a physics simulator. We used the official URDF files provided by Clearpath Robotics, which indicates the physical properties like wheel friction and slip from [41]. To train a NNDM, we first collect data on $x$, $y$, $v$, and $\theta$ as observable states from Pybullet, and use the methods in [9]. Our actions are to control linear acceleration and angular velocity, which gets converted to motor torque and the control limits are from -1 to 1. We added this information to the appendix of the revised version.
>
> > Q5: what is the rationale in having part of the code in Julia…
>
> Julia is only used to solve the SOS problem resulting from Theorem 1. This is because Julia has a matured SDP optimization tool (for SOS problems), and was the only platform capable of dealing with the optimization for all our case studies. In fact, we did initially use implementations in Python and then Matlab (via SOSTOOLS [A1]), but we opted for the Julia one due to its greater scalability for solving SOS problems. The rest of the code is in Python. In particular, for linear relaxation of NNDMs, we used the alpha-CROWN toolbox [33], and training of neural networks is in Tensorflow.
>
> [A1] Papachristodoulou et al. “SOSTOOLS - Sum of Squares Optimization Toolbox for MATLAB”, 2021
>
> > Remark 1: elaborate and provide a reference for 'called Jensen’s gap…
>
> For a convex function $B(x)$ from Jensen's inequality we get that:
>
> $E[B(x') | x] = B(E[x' | x]) + \xi,$
>
> where $\xi$ (Jensen’s gap) is a non-negative term. As a consequence, $\xi$ can be upper bounded by a non-negative polynomial function of $x$, $u$, and moments $E[v^d]$ (which for Gaussian additive noise we can compute in closed form [8, Equation 31]). Such an upper bound can be computed using [35, Equation 5].
>
> > Remark 2: l. 217 the difference between Lq(x) and Lq,x(x) is not explained…
>
> $L_q(x)$ and $L_{q,y}(x)$ are two completely different vectors of SOS polynomials. Subscripts $q$ and $y$ indicate the region and the term that the polynomial is used to bound, respectively. To reduce confusion, we changed notation $L_q(x)$ to $L_{q,x}(x)$ in the revised version.

---

> > ### Comment · Reviewer_JTrX · 2022-08-04
> > **thanks for clarifications**
> >
> > 1. I think that including the global interval bounding technique baseline is sufficient.
> > 2. ok, I get it now that the analytical models were not used during the steps of the algorithm,  however, it still makes sense to include a higher dimensional example reaching beyond 4D to see how far you can go with scalability of the approach.
> > 3. ok, this is fine.
> > 4. ok, I had in mind including exactly this kind of presentation.
> > 5. sure, definitely better to use  Julia in place of a commercial and closed source software. What is your experience in interfacing Python with Julia , smooth ?
> >
> > One question was left without comment though.
> >
> > Do you have in mind any ways of  circumventing the restrictive discretization step? This will be crucial for working out a higher dimensional example.

---

> > > ### Author Response · Authors · 2022-08-06
> > > **mitigating the discretization step and scalability**
> > >
> > > ​​We thank the reviewer for the prompt reply.
> > >
> > > > Do you have in mind any ways of circumventing the restrictive discretization step? This will be crucial for working out a higher dimensional example.
> > >
> > > We apologize for missing such a crucial point in our response. NNDMs are non-convex models, consequently completely circumventing the discretization step to formulate the optimization problem in Theorem 1 will not be easy, if possible at all in general. Nevertheless, there are various strategies we can rely on to mitigate the curse of dimensionality and scale to higher dimensions:
> > >
> > > a) we can observe that the discretization step is only needed to partition the state space in regions where the NNDM can be effectively under and over-approximated by convex (linear) functions. Therefore, we can use the distance between these bounding functions to guide the partitioning and only discretize regions where these bounds are not accurate, i.e., we split a region $R$ iff there exists $x \in R$ where the distance between upper and lower bound of $f^w(x)$ is greater than a threshold.  Note that assuming the neural network is operating close to its linear regime (as common in practice, especially for ReLU activation functions [B1 - B2]), then this approach may drastically reduce the number of required partitions. Furthermore, similar adaptive partitioning schemes have already shown promising results to scale on relatively large stochastic systems [B3 - B5];
> > >
> > > b) we can rely on decentralized approaches to compute the barrier. In particular, we can rely on the fact that the barrier does not necessarily need to be a continuous function everywhere (it must be integrable) and treat different regions of the state space independently. This approach will allow us to parallelize the synthesis of barrier functions into smaller problems each of them is potentially discretization free.
> > >
> > > [B1] Goodfellow, Ian J., Jonathon Shlens, and Christian Szegedy. "Explaining and harnessing adversarial examples." arXiv preprint arXiv:1412.6572 (2014).
> > >
> > > [B2] Zhang, Xiao, and Dongrui Wu. "Empirical studies on the properties of linear regions in deep neural networks." arXiv preprint arXiv:2001.01072 (2020).
> > >
> > > [B3] Soudjani, Sadegh Esmaeil Zadeh, and Alessandro Abate. "Adaptive gridding for abstraction and verification of stochastic hybrid systems." 2011 Eighth International Conference on Quantitative Evaluation of SysTems. IEEE, 2011.
> > >
> > > [B4] Lahijanian, Morteza, Sean B. Andersson, and Calin Belta. "Formal verification and synthesis for discrete-time stochastic systems." IEEE Transactions on Automatic Control 60.8 (2015): 2031-2045.
> > >
> > > [B5] Dutreix, Maxence, and Samuel Coogan. "Specification-guided verification and abstraction refinement of mixed monotone stochastic systems." IEEE Transactions on Automatic Control 66.7 (2020): 2975-2990.
> > >
> > > > 2. ok, I get it now that the analytical models were not used during the steps of the algorithm, however, it still makes sense to include a higher dimensional example reaching beyond 4D to see how far you can go with scalability of the approach
> > >
> > > We agree with the reviewer and have started running experiments on higher dimensional systems. But, we should mention that, with the current implementation, scaling quickly to higher dimensional systems is challenging due to the gridding procedure, which for simplicity was assumed to be uniform in each dimension. In fact, we just considered the 6D Acrobot from the OpenAI Gym, and from the first preliminary results (with a relatively coarse grid), we could not obtain any non-trivial lower bound of safety. We will continue running experiments with finer grids in order to push our current implementation to its limit and will post the final results as soon as we have them.
> > >
> > > > 5. sure, definitely better to use Julia in place of a commercial and closed source software. What is your experience in interfacing Python with Julia , smooth ?
> > >
> > > We compute the linear approximation using the Alpha-CROWN [33] toolbox in python and write them to a file. When we initialize the optimization problem (line 2 of Algorithm 1), the linear approximations are loaded from the file into Julia. As reading from the file takes a negligible amount of time, the resulting process was quite smooth.

---

> > > > ### Comment · Reviewer_JTrX · 2022-08-08
> > > > **Final remarks**
> > > >
> > > > Thank you for running the experiment on a higher dimensional example. Regarding overcoming of the curse of dimensionality, I like especially point b) leveraging the lack of necessity of global smoothness and computing the barrier in parallel on the discretized grid.   You may consider including such discussion in the paper revision as this is important to provide the reader a path towards overcoming the most important limitation of the work.
> > > >
> > > > I just noticed that you might also need to correct some of the references provided in the paper, e.g., the reference
> > > > "Ian J Goodfellow, Jonathon Shlens, and Christian Szegedy. Explaining and harnessing adversarial examples. arXiv preprint arXiv:1412.6572, 2014."
> > > > Appeared on ICLR 2015, and the published version should be referenced. This may be true about many more references (I haven't checked all references in detail).

---

> > > > > ### Author Response · Authors · 2022-08-09
> > > > > **Results for 5D and 6D benchmarks**
> > > > >
> > > > > We thank the reviewer for their reply.
> > > > >
> > > > > > Results for higher dimensional benchmarks
> > > > >
> > > > > We would like to report the results of the higher dimensional examples we have obtained so far. We considered two additional benchmarks: a 5D Husky model and the 6D Acrobat from the Open AI gym. For both the case-studies, we consider NNDM architectures with 1 hidden layer and 512 neurons. The input and output neurons are 5 and 6 for the 5D Husky and 6D Acrobot, respectively. The setup and results are reported below.
> > > > >
> > > > > *Setup:*
> > > > > - 5D Husky: A NNDM is trained by collecting the data on position $x$ and $y$, orientation $\theta$, linear and angular velocities $v$ and $w$ from Pybullet, and use the methods in [9]. We use the same approach as the 4D Husky to train a NNDM. We control linear and angular acceleration and the control limits are from -1 to 1. The state space is $x \in [-0.5, 2]$, $y \in [-0.5, 0.5]$, $\theta \in [\pi/18, \pi/18]$, $v \in [-0.5, 0.5]$, and $w \in [-0.5, 0.5]$ and the task for the robot is to stay within a lane. Thus, the safe set is defined over $y$ to be $[-0.5, 0.5]$ with the initial set being the set of all states whose $x$ and $y$ components are within the radius of 0.1 from the origin.
> > > > >
> > > > > - 6D Acrobot: We train a NNDM to imitate OpenAI Gym Acrobot model under a given expert controller. The setting we consider is same as in [R1]. It is an underactuate agent (double pendulum) with control applied to the second joint. The state space of the system is $\cos(\theta_1) \in [-0.1, 0.1]$, $\sin(\theta_1) \in [-0.6, 0.6]$, $\cos(\theta_2) \in [-0.1, 0.1]$, $\sin(\theta_2) \in [-0.6, 0.6]$, $\dot{\theta_1} \in [-0.25, 0.25]$, and $\dot{\theta_2} \in [-0.25, 0.25]$. Here $\theta_1$ is the angle of the first joint and $\theta_2$ is the angle relative to the angle of the first link.
> > > > > The task for this agent is for the tip of the second link to reach a height of y = 1. We add a safety constraint that the tip of the second link should not go beyond 1.2 and thus define the safe set to be  $\sin(\theta_1) \in [-0.6, 0.6]$ and $\sin(\theta_2) \in [-0.6, 0.6]$. Note, the maximum height the pendulum can reach is 1.8 m. Finally, the initial set is defined to be any initial point within a radius of 0.1 around the origin in the first 4 dimensions.
> > > > >
> > > > >
> > > > > *Results:*\
> > > > > 5D Husky with linear bounds:
> > > > > - |Q| = 432 (Verification): $\beta = 1.00, \quad P_s = 0.00, \quad Time = 11.64$min
> > > > > - |Q| = 432 (Control) $\quad$ : $\beta = 0.10, \quad P_s = 0.85, \quad Time = 8.34$min
> > > > > - |Q| = 1080 (Verification): $\beta = 1.00, \quad P_s = 0.00, \quad Time = 57.65$min
> > > > > - |Q| = 1080 (Control) $\quad$ : $\beta = 0.01, \quad P_s = 0.95, \quad Time = 52.08$min
> > > > > - |Q| = 1728 (Verification): $\beta = 1.00, \quad P_s = 0.00, \quad Time = 171.23$min
> > > > > - |Q| = 1728 (Control) $\quad$ : $\beta = 0.001, \quad P_s = 0.95, \quad Time = 163.80$min
> > > > >
> > > > > Acrobot 6D with linear bounds:
> > > > > - |Q| = 144 (Verification): $\beta = 1.00, \quad P_s = 0.00, \quad Time = 12.37$min
> > > > > - |Q| = 144 (Control) $\quad$ : $\beta = 10^{-6}, \quad P_s = 0.85, \quad Time = 4.45$min
> > > > > - |Q| = 288 (Verification): $\beta = 0.87, \quad P_s = 0.13, \quad Time = 26.80$min
> > > > > - |Q| = 288 (Control) $\quad$ : $\beta = 0.01, \quad P_s = 0.95, \quad Time = 12.06$min
> > > > >
> > > > > Accordingly, we will add these results to the paper, along with the global interval bounds results.
> > > > >
> > > > > We note that our framework is able to synthesize non-trivial barrier certificates for both benchmarks. However, it is to remark that by increasing the input dimension, the maximum number of partitions that our framework can handle decreases. This is because by increasing the input dimension the number of constraints and variables for each partition in Theorem 1 increase. As a consequence, to scale to higher dimensional systems, it will be essential to consider both approaches outlined in our previous response.
> > > > >
> > > > > [R1]: Paolo, Giuseppe, et al. "Guided Safe Shooting: model based reinforcement learning with safety constraints." arXiv preprint arXiv:2206.09743 (2022).
> > > > >
> > > > > > You may consider including such discussion in the paper revision as this is important to provide the reader a path towards overcoming the most important limitation of the work.
> > > > >
> > > > > We absolutely agree and will add this discussion to the paper.
> > > > >
> > > > > > I just noticed that you might also need to correct some of the references provided in the paper
> > > > >
> > > > > We thank the reviewer for spotting this. We realized there are a few references that, by mistake, we cited the arxiv version instead of the published one. We will fix all of them.

---

### Meta-Review · Area_Chair_z9d6 · 2022-09-02

**Recommendation:** Accept
**Confidence:** Less certain

**Metareview:**

This paper introduces a new approach to synthesizing barrier certificates for the verification of systems whose dynamics are defined by a neural network. There was some confusion on the exact contribution, but it was clarified in the rebuttal that while there are many prior works that aim to verify systems with neural controllers, this is the first paper that aims to provide guarantees for systems where the dynamics themselves are modeled with a neural network. This is also connected to another concern expressed in the reviews, which is the absence of a proper baseline. As the authors pointed out, the lack of any competing systems that can handle NN dynamic models makes such a baseline comparison difficult, but nevertheless, the authors added an additional baseline as part of the rebuttal process. Overall, I agree with reviewers D3ig and JTrX that this is a technically solid paper acceptable for publication at NeurIPS.

**Award:**

No

---

### Decision · Program_Chairs · 2022-09-14

Accept